# Reviving the Autopsy for Modern Cancer Evolution Research

**DOI:** 10.3390/cancers13030409

**Published:** 2021-01-22

**Authors:** Tamsin Joy Robb, Rexson Tse, Cherie Blenkiron

**Affiliations:** 1Department of Molecular Medicine and Pathology, Faculty of Medical and Health Sciences, University of Auckland, Auckland 1051, New Zealand; c.blenkiron@auckland.ac.nz; 2Department of Forensic Pathology, LabPLUS, Auckland City Hospital, Auckland 1051, New Zealand; RexsonT@adhb.govt.nz; 3Auckland Cancer Society Research Centre, University of Auckland, Auckland 1051, New Zealand

**Keywords:** cancer evolution, heterogeneity, tissue banking, autopsy, rapid autopsy, treatment resistance

## Abstract

**Simple Summary:**

Medical autopsies have a long history, but are currently experiencing a revival in order to progress modern cancer evolution research. Research autopsies represent an unparalleled opportunity to collect large volumes of cancer tissue accessible from multiple sites in a metastatic cancer patient’s body—not usually feasible during the standard clinical course. These collections enable researchers to unravel tumour evolution and heterogeneity questions. Many institutions around the world have recognised the value of these tissues, and have established rapid cancer autopsy programmes. Our article discusses a comprehensive collection of 24 rapid cancer autopsy programmes from across the globe.

**Abstract:**

Outstanding questions plaguing oncologists, centred around tumour evolution and heterogeneity, include the development of treatment resistance, immune evasion, and optimal drug targeting strategies. Such questions are difficult to study in limited cancer tissues collected during a patient’s routine clinical care, and may be better investigated in the breadth of cancer tissues that may be permissible to collect during autopsies. We are starting to better understand key tumour evolution challenges based on advances facilitated by autopsy studies completed to date. This review article explores the great progress in understanding that cancer tissues collected at autopsy have already enabled, including the shared origin of metastatic cells, the importance of early whole-genome doubling events for amplifying genes needed for tumour survival, and the creation of a wealth of tissue resources powered to answer future questions, including patient-derived xenografts, cell lines, and a wide range of banked tissues. We also highlight the future role of these programmes in advancing our understanding of cancer evolution. The research autopsy provides a special opportunity for cancer patients to give the ultimate gift—to selflessly donate their tissues towards better cancer care.

## 1. Introduction

The autopsy is one of the oldest forms of medical examination, and is the foundation for the study of human anatomy and pathology. Despite substantial modern medical advancement, particularly in medical diagnostics, the autopsy is still applicable and performed to this day. The autopsy is currently experiencing a revival in modern medical research, especially in the field of cancer evolution. This review provides an overview on the state of modern research autopsy programmes, with a focus on their potential for advancing oncology research.

## 2. A Brief History of the Autopsy and Its Role in Modern Medicine

The autopsy (literally meaning “to see for one’s self”) is synonymous with post mortem examination and necropsy [1]. It is a specialised medical examination of deceased bodies performed by qualified pathologists, which has long held great value in the medical and scientific community.

The modern autopsy practice originates from Europe, where it was initially used for medico-legal purposes [2]. From the 13th century onwards, the autopsy was gradually recognised for its use in medical studies. The 16th century (the Renaissance) was described as the “Golden Age of Anatomy”, in which the purpose of the autopsy shifted to encompass anatomical dissection for medical learning [3,4]. In the second half of the 18th century, to build on the understanding of the anatomy, the autopsy was applied to the study of pathology [5]. The autopsy subsequently made significant contributions to the early understanding of disease processes; for example, scientists were able to prove that human disease followed a traceable cause and effect, refuting the opposing theory of the time that disease was instead governed by spiritual beings [5,6]. Pioneering work on autopsy specimens contributed mounting evidence that diseases originated at and were visible at a cellular level [7]. Autopsy studies also helped shape clinical practice in the early 1900s [8], helping to reduce incorrect diagnoses.

The modern autopsy techniques were developed by Friedrich Albert Zenker and Rudolf Virchow in the 19th Century [7,9]. Zenker and Virchow had contrasting approaches to autopsy; where Zenker would remove organs in “blocks” in which the organs were connected to each other, Virchow removed and examined single organs. These two different techniques (and their variations), together with the use of the microscope, provide the foundations upon which the modern day autopsy is built [7,10,11]. For a fascinating detailed review of the history of the autopsy, see that authored by Cecchetto et al. [10].

In present times, the autopsy is a specialized medical examination that encompasses more than just dissection. It includes reviewing medical records, post mortem radiology (X-ray, CT, and MRI), external and internal (dissection) examination, microscopic/histological examination, and ancillary tests (genetics, biochemistry, immunology, microbiology, and toxicology) [10,11]. The autopsy enables the entire body to be examined using multiple modalities (radiology, external inspection, and internal dissection) and allows access to all tissue types, enabling them to be collected for analysis, including those not easily accessible during life. It is termed as “the ultimate”, and thus final medical examination a person can receive. Autopsies are performed in both hospital and forensic settings, servicing the medical and coronial systems, respectively. The purpose of autopsies performed in these two settings are different-within the medical system, autopsies contribute to medical education, audits, and research, whereas forensic/coronial autopsies aid in death investigations for medico-legal reasons.

Although forensic/coronial autopsies are mildly increasing in number from the recent drug abuse epidemic [12,13], the request for medical autopsies has declined significantly because of the advancement in medical diagnostics, with fewer than 5% of hospital deaths now being autopsied [14]. In pathology, most research is now conducted at cellular and genetic levels, rather than investigating the gross morphology. Medical research has therefore largely moved away from the use of the autopsy [3]. Despite this, scientists and clinicians have recently begun reinvigorating autopsy programmes for use in medical research, as medical autopsies can provide unique research materials through extensive tissue access; a powerful opportunity to advance our understanding of disease [15]. Examples include Huntington’s and Alzheimer’s disease [16,17,18], multiple sclerosis [19], and the focus of this article-cancer [15,20]. The following sections examine the use of medical autopsies for the collection and storage of cancer tissue samples to facilitate answering pressing clinical questions in modern cancer research, with a focus on contributions improving our understanding of cancer evolution and metastasis.

## 3. What Is a Research Autopsy Programme

The role of autopsy for cancer research is in the collection of cancer tissue from deceased donors [15]. Medical institutions around the world have recognised the value of precious tumour tissues donated from patients consenting to autopsy, and have responded with funding ongoing programmes dedicated to collecting tumour tissue through autopsies [21]. Some universities or research hospitals, often in collaboration with tissue banking-targeted programs and research teams, run dedicated autopsy programmes, often involving permanent staff and facilities set up to identify and consent potential donors; collect consented autopsy tumour tissues after their death; and support downstream processing, storage, and analysis of the collected tissue. Research autopsies are often called “rapid” autopsies, because of the urgent nature of sample collection after patient death in order to preserve the molecular information within the tissues. Running a research autopsy programme can be very labour intensive and, in some cases, involves having staff available 24 h a day, 7 day a week [22,23], as well as an integrated body transfer service for patients not in hospital care [24]. In other cases, best efforts are promised to donors who pass away during shorter operating hours of a programme [25]. Similar to any medical research involving human subjects, research autopsy programmes run under a strict ethical code, with informed consent at the heart of their work [26] and are guided and made possible by generous patients and their families, strongly wishing to contribute to future cancer research [27,28,29]. Furthermore, as the programme involves deceased individuals, it must also adhere to local coroner’s and human tissue acts [30,31].

## 4. What Can the Autopsy Provide to Modern Cancer Research

The utility of tumour samples collected at autopsy is primarily to study important clinical problems, including metastasis, tumour evolution, and treatment resistance. Through autopsies, investigators can access, collect, and catalogue the entire tumour mass (including primary and any metastatic tumours) present throughout the body—infeasible through standard tissue banking procedures in living individuals [15].

Cancer evolution is an increasingly important topic, and a search is underway for new frameworks to understand the evolutionary dynamics of cancer [32,33]. To validate different evolutionary models, it is essential to compare (1) the relationship among primary tumour and different metastatic sites (from different organs) of the same individual, and (2) the relationship of the same cancer type among different individuals. The well-suited resources of the tumour tissues collected at autopsy are thus crucial for cancer evolutionary research. The ability to study tumour evolution from tissues sampled under clinical sampling or at autopsy is compared in Figure 1.

Standard tissue banking suffers from sampling biases as it relies on tissues donated from surgical resections; thus, advanced disease not suitable or inaccessible for resection is excluded from analysis. An alternative approach to avoid spatial sampling bias is the use of small invasive biopsies collected solely for research purposes. However, this is often plagued with its own biases by virtue of the limited size of samples that are ethically permissible to collect. A summary of these differences, through the lens of suitability for studying cancer evolution, is presented in Figure 2.

Researchers need appropriate tissue samples to best explore research questions around the heterogeneous and evolving nature of cancer, particularly at a genomic level. The autopsy provides just this opportunity; cancer patients who consent to donate their tissues through an autopsy following their death are often gifting researchers access to tissues considered clinically and ethically un-obtainable during life, in greater quantities, and from a greater number of sites per patient in the case of highly disseminated disease (sometimes even including sampling of lesions not discernible from imaging scans). Carefully designed autopsy programmes provide researchers with the opportunity to ensure that the samples collected are fit for their intended research purposes: designing studies around downstream applications, and customising fixation techniques. This includes, for example, collecting matched fresh snap-frozen and formalin fixed samples to study the cancer genomics and morphology concurrently [21,24]. This protocol design enables researchers to simultaneously and precisely compare multiple cancer samples, within a single patient and across groups of patients, at one time point.

Depending on the informed consent provided, and on the country legislation, autopsy programmes can facilitate in the collection of a wide range of body tissues relevant to cancer research. This includes soft tissue, bony tissue, organs adjacent, or distant to the cancer, as well as blood and other body fluids [34,35]. These samples are collected for a wide range of downstream applications, as determined by the research group, including genomics (including analysing DNA, RNA, and methylation), histopathology, single cell analyses, and the establishment of cell lines or patient-derived xenografts [36], and, in some cases, all of the above [25].

Despite the clear advantages compared with clinical sampling, cancer samples collected at autopsy have notable limitations. Samples collected at autopsy represent a single time-point of end-stage disease, limiting their ability to be used to track temporal and treatment-induced changes. To complement autopsy tissues, some studies also have access to matched surgical samples collected during the patient’s clinical care [37,38], a valuable combination that allows researchers to investigate treatment-related changes, and better time evolutionary events. A further limitation is in the size of the tumour it is possible to collect: tumour samples collected at autopsy are relatively large (at least > 1–2 mm), which are visible by the naked eye or through imaging, thus limiting the procurement of potential small precursor lesions, which may be too small to be detected. Collecting adjacent uninvolved tissue samples at autopsy could however be included in the study protocol, in an attempt to capture and study localised precursor lesions. In addition to these limitations, samples may initially be “banked” rather than being immediately used for research; therefore, opportunities to complete downstream applications requiring the use of fresh tissues, such as single cell dissemination or xenograft development, may be lost. Although deemed to be “rapid”, extensive autopsies can be challenged by the time it takes to sample and document all sites, meaning that later-collected and fixed samples may suffer post mortem changes, an effect that varies between tissue types [39]. Finally, the extent of the sampling and the tissue types preserved are often staff- and resource-dependent. Tissue procurement from cases with widely disseminated cancer can be technically challenging; for example, sampling tumours in the vertebral column requires radiology interpretation and subsequent disarticulation of vertebral bodies. This may require the availability and expertise to interpret radiological scans, as well as the specialised technical skills of the dissector/pathologist in collecting the tissue required.

## 5. Autopsy Programmes around the World

Established autopsy programmes collecting cancer tissues for use in research are growing in number around the world; the most well-known are in North America. Prominent programmes also exist in Australia, the United Kingdom, Europe, and South America, however such programmes are not always clearly documented in the literature, so there are likely many more in existence (e.g., see Japanese studies [40,41]). Table 1 summarises these documented programs and their key scientific contributions to the field of cancer biology.

From this extensive (but not exhaustive) list of documented autopsy programmes, there is indeed an increasing global focus on collecting invaluable cancer tissue samples for research. However, not all programmes are the same; some programmes collect defined cancer types (metastatic prostate [77], breast [21], and pancreatic [71] cancer being the most common), while other programmes include a wider range of cancers. Common to all programmes is the strong tie between tissue collections and research groups. This is to ensure resources are in place to make sufficient research use of generously donated cancer tissues. While most well-documented programmes are located in the USA (past reviews of this field have focussed almost exclusively on these programmes [83,84,85]), evidence of prolific programmes outside the USA is emerging, including in Brazil [82], Spain [48], and Hungary [50], as listed in Table 1. Clearly, the value of research autopsy programmes is internationally recognised, acknowledging the significant efforts required to establish each with regards to local cultural; ethical, legal, and societal factors. The global presence of research autopsy programmes is invaluable for a plethora of reasons—it paves the way for diversity in the populations studied (and their underlying genetic makeup); diversity in cancer types (as some cancer types are more prevalent in different areas of the world); and critically, diversity in treatment regimens (when drug funding or access varies, patients’ disease may be more likely to be treatment naive versus treated with a complex sequence of drugs both scenarios are important to researchers).

A wealth of new knowledge on cancer biology has been uncovered by these programmes. Such studies have solidified the understanding of the shared origin of metastatic cells (i.e., tumours often arise from a single initial cell). The processes of cancer evolution have been investigated across a range of cancer types, providing a deeper understanding based on more samples than previously possible, and across sizeable groups of patients. Research autopsy tissues have enabled studies into cancer drug and treatment response, and have furthered our understanding of the development of treatment resistance, one of the greatest clinical problems in oncology today. We now also appreciate that low-frequency somatic cancer mutations are found in normal tissues of healthy patients [86]. Finally, research autopsies have provided viable cancer tissues to researchers for the development of new living cancer models, which future studies can be based on, including cell lines and patient derived xenografts. Contributions of selected research autopsy programmes to furthering our understanding of cancer evolution are discussed below.

## 6. Spotlight on Selected Research Autopsy Programmes

University of Michigan: The seminal study published in 2000 is credited as the first modern cancer research autopsy programme, and showed the establishment of suitable protocols to yield high quality tumour tissues suitable for generating cell lines and xenografts, and also established logistical procedures to enable round-the-clock collections [68]. This study paved the way for the many other similar programmes listed in this review, including the Weill Cornell Precision Medicine Programme, which directly credits programme directors Rubin and Pienta for their autopsy protocols [80]. The tissue collected has enabled deep investigations into castration-resistant prostate cancer in particular, providing evidence of highly recurrent mutations that may have prognostic or treatment significance [67].

Johns Hopkins: This institute initially established multiple parallel autopsy programmes targeting specific cancer streams, in particular breast, prostate (PELICAN), and gastrointestinal cancers [58]. Following the successes of these prolific targeted programmes, the inclusion criteria widened to other cancer types in 2014. This programme can be credited with many contributions to our understanding of tumour evolution and metastasis, including the clonality of primary and metastatic lesions, patterns of dissemination to metastatic sites, and even large studies on treatment-naive patients [87]. Through this tissue, we have a better understanding of the genes altered at key branching points (like propensity for metastasis or development of treatment resistance) in tumour evolution. A wide assortment of patient derived xenografts and cell lines provides ongoing resources to support a range of future research questions [60].

TRACERx study (TRAcking Cancer Evolution through therapy (Rx)): This is an ongoing prospective observational study funded by Cancer Research UK, currently mainly focussed on renal [38] and lung cancers [88]. In conjunction with the PEACE (Posthumous Evaluation of Advanced Cancer Environment) study [47], researchers are able to follow patients through their lives, including collecting metastatic biopsies and regular blood samples, as well as consenting patients to donate their tumour tissues through rapid autopsy after their death. Combining a prospective study with autopsy collections adds power to their findings, and is an approach that should be adopted by other programmes, where resources allow; it enables advancing our understanding of evolution across time, and provides a framework for answering questions around treatment resistance and other drug-related changes, the response of the immune system, and temporal tumour heterogeneity. This study was the focus of a recent *Nature* collection, highlighting the breakthroughs that this programme has revealed. Insights include the importance of whole-genome doubling events early on in tumour development, which appears to amplify the genes needed for tumour survival [89], as well as detailed work investigating the genome sequences of T-cell receptors, which revealed that they evolve in parallel with the tumour, but eventually get outpaced as the tumour evades the immune system through chromosomal instability [90,91]. Going forward, tissues obtained through this autopsy programme may be the key to critical unanswered questions, including how cancer cells spread from the primary tumour to distant sites (including the role of cells shed during surgical interventions) [88].

*n* = 1 Research Autopsy Programmes: Where no dedicated programmes exist, some research teams are turning to innovative measures to meet patient wishes and enable the acceptance of generous patient tissue donations, by constructing *n* = 1 research autopsy programmes [81]. One such example of this is our own *n* = 1 research autopsy programme in Auckland, New Zealand, where no routine research autopsy programme for cancer patients exists [92]. Despite the considerable effort required to construct an *n* = 1 programme, the ability to tailor the programme to the particular needs of the patient, their disease, and the research question, has clear advantages over a higher throughput programme approach. Designing this programme encompassed local ethical, legal, and cultural considerations, logistics, and standard operating procedures best suited to the tissue samples collected. *n* = 1 programmes such as this would not be possible without the insights gained from large and long-standing institutional research autopsy programmes, but perhaps in turn the insights from bespoke *n* = 1 studies may be applied to larger programmes. The ability to customise the approach to the tissue collection, to include a rare cancer type not usually the subject of targeted collections, to collect a wide range and breadth of samples (over 400 samples in our *n* = 1 study compared to a maximum of 24 from a single patient in Johns Hopkins’ routine programme [93], for example), and to meticulously complete point of collection spatial mapping to reconstruct the relative positions of these samples during the analysis phase, is particularly valuable.

## 7. Conclusions

The value of metastatic cancer tissue collected at autopsy is clear, and is clearly recognised internationally by research institutions hosting research autopsy programmes. Such programmes have enabled the collection of tissues usually inaccessible to researchers, in greater quantities than otherwise available, and in optimal condition for modern genomic analyses, the generation of cell lines and xenografts, and other future research applications. This tissue has in turn generated its own wealth of scientific breakthroughs in our understanding of how specific cancer types, including breast, prostate, pancreatic, lung, and even difficult to study rare cancer types, spread through the body, seed metastases, and develop resistance to treatment-some of the greatest challenges facing oncologists and their patients today. Tumour samples collected at autopsy represent the ultimate selfless donation by cancer patients, and the substantial efforts of highly organised research autopsy programmes are truly honouring these donations. In the words of Johns Hopkins Rapid Autopsy Programme Director, Dr. Jody Hooper, “These patients are passing the torch to us. They have had their fight with cancer. We can carry on their fight, even after they are gone” [94].

## Figures and Tables

**Figure 1 cancers-13-00409-f001:**
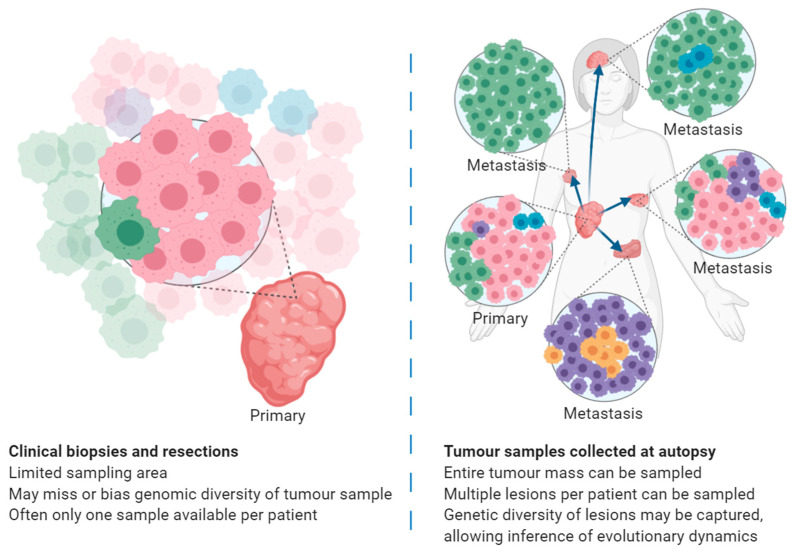
Demonstration of tumour heterogeneity as captured by small clinical biopsies and resections (**left**) compared with tumour samples collected at autopsy (**right**). Cancer cells are coloured by genomic similarity, where cancer cells of the same colour share the same set of genomic changes. In small cancer samples, such as those collected during clinical biopsy and resection procedures, the sample may or may not capture all genetic variations present in that tumour at that time point, and provides no information about other tumours around a patient’s body. Comparatively, tumour samples collected at autopsy may encompass larger or complete tumours and cover the primary tumour and all metastases present in the patient, allowing for genomic analyses comparing the genomic changes present in each sample. To validate different evolutionary models and patterns, it is essential to study multiple tumour samples from tumour sites around a patient’s body, usually only feasible on tumour samples collected at autopsy.

**Figure 2 cancers-13-00409-f002:**
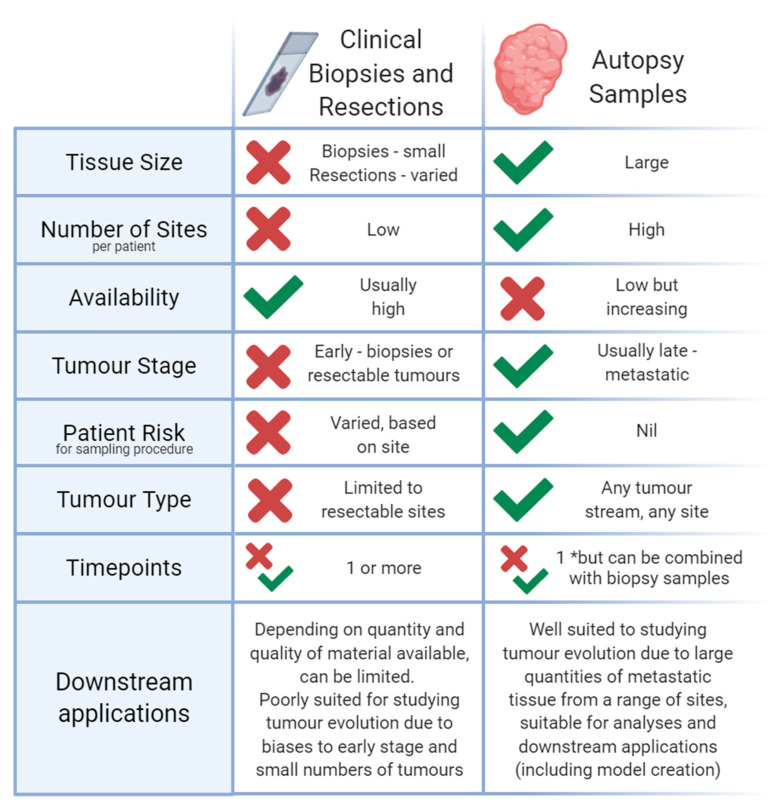
Comparison of tissue samples collected through clinical care (including biopsies and resections) with those collected at autopsy after the patient’s death. Autopsy samples are better suited to answering many questions in cancer evolution than those collected through clinical care, because of the availability of larger volumes of metastatic tissue from a variety of sites from the same patient, however clinical samples are more readily available to researchers in many parts of the world at this time (although the increase in autopsy programmes is notable), and provide information on different timepoints of disease, including before and after treatment, and are thus particularly valuable when used alongside autopsy tissues in cancer evolution studies.

**Table 1 cancers-13-00409-t001:** Autopsy programmes and their contribution to cancer biology.

Continent	Programme Name, Host Institution	Tumour Types	Contributions
Australasia	CASCADE-Peter MacCallum Cancer Centre, Melbourne, Australia	All (with a focus on metastatic breast cancer)	Strong community angle, with a focus on molecular heterogeneity in breast cancer, highlighting heterogeneity in subclones, parallel evolution of treatment resistance, and metastatic cross-seeding [21,24,42].
	Royal Brisbane and Women’s Hospital, Brisbane, Australia	Metastatic breast cancer	Long-standing programme, in operation for over 50 years [35,43].
Asia	Japan, “Liquid Autopsy” proof of concept study, Akita Rapid Autopsy Program	Prostate cancer	This study proposes the value of alternative “liquid autopsies” to study cancer, overcoming the labour-intensive and expensive drawbacks of tissue programmes; particularly suggesting this approach for smaller institutions without the resources for large-scale autopsy tissue programmes [34,44].
UK and Europe	University of Cambridge, UK	Metastatic breast cancer and other	Seminal study linking *n* = 1 autopsy samples to changes seen in the plasma, in relation to differing treatment responses across metastatic sites [45,46].
	University College London, PEACE (Posthumous Evaluation of Advanced Cancer Environment), UK	Renal and lung cancer	Large-scale renal and lung cancer post-mortem studies [47], tied to prolific TRACERx studies [37,38] (see SPOTLIGHT below).
	Spanish National Cancer Research Centre (CNIO), Madrid, Spain	Pancreatic cancer	Molecular tumour evolution investigation through xenograft studies [48].
	Vall D’Hebron Institute of Oncology Warm Autopsy Program, Barcelona, Spain	All	This autopsy programme contributed to the understanding of therapy-resistant metastatic breast cancer, revealing patterns of evolution resembling communities of clones, accumulation of HLA loss of heterogeneity, and variable tumour microenvironments [46,49].
	Second Department of Pathology at Semmelweis University, Budapest, Hungary	All, with a focus on breast cancer	An example of a strong addition to the field coming from outside traditional centres, completing 80 routine cancer autopsies annually, with contributions to evolutionary modelling of cancer data [50].
North America	Pan-Cancer Research Autopsy Programme, Princess Margaret Cancer Centre, University Health Network, Toronto, Canada	Pan-cancer	A productive pan-cancer programme that has completed over 100 cancer autopsies to date, with a strong focus on pre-autopsy planning in consultation with research scientists, in order to capture identified treatment responsive and resistant lesions for genomic study [51]. This programme has the potential to advance our knowledge of a large range of cancers, including more rare subtypes that may otherwise be excluded from existing targeted programmes.
	Brain and Body Donation Program (BBDP), Banner Sun Health Research Institute	All	Long-standing brain (1987-) and cancer bank (2005-), a well-staffed “consistently-rapid” autopsy programme [22,52,53].
	Johns Hopkins	Initially prostate (PELICAN), breast, and pancreatic (Gastrointestinal Cancer Rapid Medical Donation program), but from 2014 onwards all tumour types	Improved our understanding of genetic and phenotypic heterogeneity, including biomarker heterogeneity, between metastatic sites. Vast and highly successful programme responsible for many of the seminal papers in tumour evolution and heterogeneity field [54,55,56,57,58,59,60,61,62] (see SPOTLIGHT below).
	Massachusetts General Hospital Cancer Center	All tumour types	Strong focus on “patient avatars” [36].
	Memorial Sloan Kettering Cancer Center (Last Wish Program)	All tumour types	Multimodal evolutionary studies integrating histological and genomic data [63,64]. MSK have conducted extensive interviews of stakeholders involved in process to garner perspectives on tissue donation [29].
	Moffitt Cancer Center Rapid Tissue Donation Program	Lung cancer	An example of a strong consultation of stakeholders in developing the autopsy programme, providing guidance to other centres [65].
	National Cancer Institute (NCI)	Lung cancer	Integrated transcriptomic and proteogenomic rapid autopsy programme advancing the understanding of tumour evolution and heterogeneity [66].
	Ohio State University Comprehensive Cancer Center	All	Established programme completing 12 cases per year, with a focus on developing new analysis tools and generating cell lines [25].
	University of Michigan	All cancer types	This is a seminal programme, focussing on metastatic prostate cancer, which forms the basis of so many other programmes [67,68,69,70] (see SPOTLIGHT below).
	University of Nebraska Medical Center	Pancreatic cancer	This institute employs a large permanent autopsy team focussing on high throughput and quality rapid autopsy sample collection, enabling high resolution tumour evolution studies. They have performed over 100 autopsies to date [71].
	University of North Carolina Chapel Hill	Breast cancer	Genomics-enabled tumour evolution autopsy studies, contributing to our understanding of patterns of metastatic seeding in breast cancer [72,73].
	University of Pittsburgh	Focus on prostate, breast, and lung cancer	Strong focus on learning lessons from patients to build a strong and productive programme [27]. It is also considered a seminal programme that has shaped many other current programmes [74].
	University of Utah	Breast cancer	Core focus on the development of bioinformatic tools for analysing multiple samples [75].
	University of Washington Medical Center Cancer Donor Autopsy Program	Prostate cancer and urothelial cancer	Long-standing prostate cancer programme that began in 1989 [76,77,78,79].
	Weill Cornell Precision Medicine Program/Rapid Autopsy Program	Metastatic prostate and other select types.	Strong focus on “next generation rapid autopsies” generating genomic data, and strong proponents of providing cancer autopsies [80,81].
South America	Sao Paulo, Brazil	Instituto do Câncer do Estado de São Paulo (ICESP)	A key established programme outside North American research institutions, which has been running 24 h a day since 1980, completing approximately 60 autopsies a year [23,82]. Some samples are collected under prospective studies, in which case customisations are made to tissue sampling protocols. Interestingly, Brazilian legislation requires pathologists to wait for at least 6 h after death to perform an autopsy, which impacts the fragmentation of RNA, and to a lesser extent DNA; however, the programme must operate within the legal framework of Brazil.

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
