# Peer review of "Reviving the Autopsy for Modern Cancer Evolution Research"

_cancers, 2021, doi:10.3390/cancers13030409_

Round 1

Reviewer 1 Report

I read with pleasure this manuscript focused on the importance of autopsy in modern era.

It is a review paper, presenting the state of art and future prospects on autopsies.

The concepts are clearly presented and supported by accurate literature review.

The possible clinical significance of performing autopsies programmes on metastatic cancer can be difficult to understand, but surely the definition of the molecular biology underlying metastastic spread can pave the way for future translational studies.

Author Response

Dear reviewer, 

Thank you for your positive comments on our manuscript.

Reviewer 2 Report

The review manuscript by Robb, Tse, and Blenkiron entitled “Reviving the autopsy for modern cancer evolution research” provides a historical overview of the autopsy, discusses the current state of the autopsy programs and future application of the autopsy for analysis of cancer evolution and heterogeneity. The paper is timely, comprehensive, and very well written, and could be of interest to the broad range of readers interested in cancer research.

Comment:
To improve the clarity and make this work even more interesting for the readers of different disciplines, I would suggest including additional figure giving examples of how cancer evolution and heterogeneity can be analyzed in autopsy samples with possible links to the current autopsy programs.

Author Response

Dear reviewer, 
Thank you for your positive comments on our manuscript. We appreciated your suggestion of an additional figure, and have now included this to show visually the difference between samples collected through clinical biopsies or at autopsy, in terms of capturing tumour heterogeneity to study evolution. It is included as the new Figure 1, and the original figure is now Figure 2. 

Many thanks for your helpful suggestion.